# META LEARNING SHARED HIERARCHIES

**Kevin Frans**
Henry M. Gunn High School
Work done as an intern at OpenAI
kevinfrans2@gmail.com

**Jonathan Ho, Xi Chen, Pieter Abbeel**
UC Berkeley, Department of Electrical
Engineering and Computer Science

**John Schulman**
OpenAI

## ABSTRACT

We develop a metalearning approach for learning hierarchically structured policies, improving sample efficiency on unseen tasks through the use of shared primitives—policies that are executed for large numbers of timesteps. Specifically, a set of primitives are shared within a distribution of tasks, and are switched between by task-specific policies. We provide a concrete metric for measuring the strength of such hierarchies, leading to an optimization problem for quickly reaching high reward on unseen tasks. We then present an algorithm to solve this problem end-to-end through the use of any off-the-shelf reinforcement learning method, by repeatedly sampling new tasks and resetting task-specific policies. We successfully discover[1] meaningful motor primitives for the directional movement of four-legged robots, solely by interacting with distributions of mazes. We also demonstrate the transferability of primitives to solve long-timescale sparse-reward obstacle courses, and we enable 3D humanoid robots to robustly walk and crawl with the same policy.

## 1 INTRODUCTION

Humans encounter a wide variety of tasks throughout their lives and utilize prior knowledge to master new tasks quickly. In contrast, reinforcement learning algorithms are typically used to solve each task independently and from scratch, and they require far more experience than humans. While a large body of research seeks to improve the sample efficiency of reinforcement learning algorithms, there is a limit to learning speed in the absence of prior knowledge.

We consider the setting where agents solve distributions of related tasks, with the goal of learning new tasks quickly. One challenge is that while we want to share information between the different tasks, these tasks have different optimal policies, so it's suboptimal to learn a single shared policy for all tasks. Addressing this challenge, we propose a model containing a set of shared sub-policies (i.e., motor primitives), which are switched between by task-specific master policies. This design is closely related to the options framework (Sutton et al., 1999; Bacon et al., 2016), but applied to the setting of a task distribution. We propose a method for the end-to-end training of sub-policies that allow for quick learning on new tasks, handled solely by learning a master policy.

Our contributions are as follows.

- We formulate an optimization problem that answers the question of *what is a good hierarchy?*—the problem is to find a set of low-level motor primitives that enable the high-level master policy to be learned quickly.

- We propose an optimization algorithm that tractably and approximately solves the optimization problem we posed. The main novelty is in how we repeatedly reset the master policy, which allows us to adapt the sub-policies for fast learning.

---

[1]Videos at https://sites.google.com/site/mlshsupplementals

We will henceforth refer to our proposed method—including the hierarchical architecture and optimization algorithm—as MLSH, for *metalearning shared hierarchies*.

We validate our approach on a wide range of environments, including 2D continuous movement, gridworld navigation, and 3D physics tasks involving the directional movement of robots. In the 3D environments, we enable humanoid robots to both walk and crawl with the same policy; and 4-legged robots to discover directional movement primitives to solve a distribution of mazes as well as sparse-reward obstacle courses. Our experiments show that our method is capable of learning meaningful sub-policies solely through interaction with a distributions of tasks, outperforming previously proposed algorithms. We also display that our method is efficient enough to learn in complex physics environments with long time horizons, and robust enough to transfer sub-policies towards otherwise unsolvable sparse-reward tasks.

## 2 RELATED WORK

Previous work in hierarchical reinforcement learning seeks to speed up the learning process by recombining a set of temporally extended primitives—the most well-known formulation is Options (Sutton et al., 1999). While the earliest work assumed that these options are given, more recent work seeks to learn them automatically (Vezhnevets et al., 2016; Daniel et al., 2016). Heess et al. (2016) discovers primitives by training over a set of simple tasks. Florensa et al. (2017) learns a master policy, where sub-policies are defined according to information-maximizing statistics. Bacon et al. (2016) introduces end-to-end learning of hierarchy through the options framework. Henderson et al. (2017) extends the options framework to include reward options. Several methods (Dayan & Hinton, 1993; Vezhnevets et al., 2017; Ghazanfari & Taylor, 2017) aim to learn a decomposition of complicated tasks into sub-goals. These prior works are mostly focused on the single-task setting and don't account for the multi-task structure as part of the algorithm. Other past works (Thomas & Barto, 2011; Thomas, 2011; Thomas & Barto, 2012) have simultaneously learned modules that are used in conjunction to solve tasks, but do not incorporate temporal abstraction. On the other hand, our work takes advantage of the multi-task setting as a way to learn temporally extended primitives.

There has also been work in metalearning, where information from past experiences is used to learn quickly on specific tasks. Andrychowicz et al. (2016) proposes the use of a recurrent LSTM network to generate parameter updates. Duan et al. (2016) and Wang et al. (2016) aim to use recurrent networks as the entire learning process, giving the network the same inputs a traditional RL method would receive. Mishra et al. (2017) tackles a similar problem, utilizing temporal convolutions rather than recurrency. Finn et al. (2017) accounts for fine-tuning of a shared policy, by optimizing through a second gradient step. While the prior work on metalearning optimizes to learn as much as possible in a small number of gradient updates, MLSH (our method) optimizes to learn quickly over a large number of policy gradient updates in the RL setting—a regime not yet explored by prior work.

## 3 PROBLEM STATEMENT

First, we will formally define the optimization problem we would like to solve, in which we have a distribution over tasks, and we would like to find parameters that enable an agent to learn quickly on tasks sampled from this distribution.

Let $S$ and $A$ denote the state space and action space, respectively. A Markov Decision Process (MDP) is defined by the transition function $P(s', r|s, a)$, where $(s', r)$ are the next state and reward, and $(s, a)$ are the state and action.

Let $P_M$ denote a distribution over MDPs $M$ with the same state-action space $(S, A)$. An *agent* is a function mapping from a multi-episode history $(s_0, a_0, r_0, s_1, a_2, r_2, \ldots s_{t-1})$ to the next action $a_t$. Specifically, an agent consists of a reinforcement learning algorithm which iteratively updates a parameter vector $(\phi, \theta)$ that defines a stochastic policy $\pi_{\phi,\theta}(a|s)$. $\phi$ parameters are shared between all tasks and held fixed at test time. $\theta$ is learned from scratch (from a zero or random initialization) per-task, and encodes the state of the learning process on that task. In the setting we consider, first an MDP $M$ is sampled from $P_M$, then an agent is incarnated with the shared parameters $\phi$, along with randomly-initialized $\theta$ parameters. During an agent's $T$-step interaction with the sampled MDP $M$, the agent iteratively updates its $\theta$ parameters.

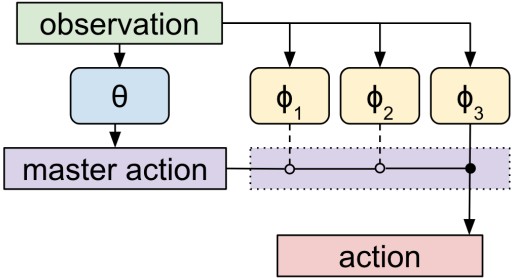

Figure 1: Structure of a hierarchical sub-policy agent. $\theta$ represents the master policy, which selects a sub-policy to be active. In the diagram, $\phi_3$ is the active sub-policy, and actions are taken according to its output.

In other words, $\phi$ represents a set of parameters that is shared between tasks, and $\theta$ represents a set of per-task parameters, which is updated as the agent learns about the current task $M$. An agent interacts with the task for $T$ timesteps, over multiple episodes, and receives total return $R = r_0 + r_1 + ... + r_{T-1}$. The meta-learning objective is to optimize the expected return during an agent's entire lifetime, over the sampled tasks.

$$\text{maximize}_\phi \; E_{M \sim P_M, t=0...T-1}[R] \tag{1}$$

This objective tries to find a shared parameter vector $\phi$ that ensures that, when faced with a new MDP, the agent achieves high $T$ time-step returns by simply adapting $\theta$ while in this new MDP.

While there are various possible architectures incorporating shared parameters $\phi$ and per-task parameters $\theta$, we propose an architecture that is motivated by the ideas of hierarchical reinforcement learning. Specifically, the shared parameter vector $\phi$ consists of a set of subvectors $\phi_1, \phi_2, \ldots, \phi_K$, where each subvector $\phi_k$ defines a sub-policy $\pi_{\phi_k}(a|s)$. The parameter $\theta$ is a separate neural network that switches between the sub-policies. That is, $\theta$ parametrizes a stochastic policy, called the *master policy* whose action is to choose the index $k \in \{1, 2, \ldots, K\}$. Furthermore, as in some other hierarchical policy architectures (e.g. options (Sutton et al., 1999)), the master policy chooses actions at a slower timescale than the sub-policies $\phi_k$. In this work, the master policy samples actions at a fixed frequency of $N$ timesteps, i.e., at $t = 0, N, 2N, \ldots$.

This architecture is illustrated in Figure 1. By discovering a strong set of sub-policies $\phi$, learning on new tasks can be handled solely by updating the master policy $\theta$. Furthermore, since the master policy chooses actions only every $N$ time steps, it sees a learning problem with a horizon that is only $1/N$ times as long. Hence, it can adapt quickly to a new MDP $M$, which is required by the learning objective (Equation (1)).

## 4 Algorithm

We would like to iteratively learn a set of sub-policies that allow newly incarnated agents to achieve maximum reward over $T$-step interactions in a distribution of tasks.

An optimal set of sub-policies must be fine-tuned enough to achieve high performance. At the same time, they must be robust enough to work on wide ranges of tasks. Optimal sets of sub-policies must also be diversely structured such that master policies can be learned quickly. We present an update scheme of sub-policy parameters $\phi$ leading naturally to these qualities.

### 4.1 Policy Update in MLSH

In this section, we will describe the MLSH (metalearning shared hierarchies) algorithm for learning sub-policy parameters $\phi$. Starting from a random initialization, the algorithm (Algorithm 1) iteratively performs update steps which can be broken into two main components: a warmup period to optimize master policy parameters $\theta$, along with a joint update period where both $\theta$ and $\phi$ are optimized.

---

**Algorithm 1** Meta Learning Shared Hierarchies

Initialize $\phi$
**repeat**
    Initialize $\theta$
    Sample task $M \sim P_M$
    **for** $w = 0, 1, ...W$ (warmup period) **do**
        Collect $D$ timesteps of experience using $\pi_{\phi,\theta}$
        Update $\theta$ to maximize expected return from $1/N$ timescale viewpoint
    **end for**
    **for** $u = 0, 1, ....U$ (joint update period) **do**
        Collect $D$ timesteps of experience using $\pi_{\phi,\theta}$
        Update $\theta$ to maximize expected return from $1/N$ timescale viewpoint
        Update $\phi$ to maximize expected return from full timescale viewpoint
    **end for**
**until** convergence

---

From a high-level view, an MLSH update is structured as follows. We first sample a task $M$ from the distribution $P_M$. We then initialize an agent, using a previous set of sub-policies, parameterized by $\phi$, and a master policy with randomly-initialized parameters $\theta$. We then run a warmup period to optimize $\theta$. At this point, our agent contains of a set of general sub-policies $\phi$, as well as a master policy $\theta$ fine-tuned to the task at hand. We enter the joint update period, where both $\theta$ and $\phi$ are updated. Finally, we sample a new task, reset $\theta$, and repeat.

The warmup period for optimizing the master policy $\theta$ is defined as follows. We assume a constant set of sub-policies as parameterized by $\phi$. From the sampled task, we record $D$ timesteps of experience using $\pi_{\phi,\theta}(a|s)$. We view this experience from the perspective of the master policy, as in Figure 2. Specifically, we consider the selection of a sub-policy as a single action. The next $N$ timesteps, along with corresponding state changes and rewards, are viewed as a single environment transition. We then update $\theta$ towards maximizing reward, using the collected experience along with an arbitrary reinforcement learning algorithm (for example DQN, A3C, TRPO, PPO) (Mnih et al., 2015; 2016; Schulman et al., 2015; 2017). We repeat this prodecure $W$ times.

Next, we will define a joint update period where both sub-policies $\phi$ and master policy $\theta$ are updated. For $U$ iterations, we collect experience and optimize $\theta$ as defined in the warmup period. Additionally, we reuse the same experience, but viewed from the perspective of the sub-policies. We treat the master policy as an extension of the environment. Specifically, we consider the master policy's decision as a discrete portion of the environment's observation. For each $N$-timestep slice of experience, we only update the parameters of the sub-policy that had been activated by the master policy.

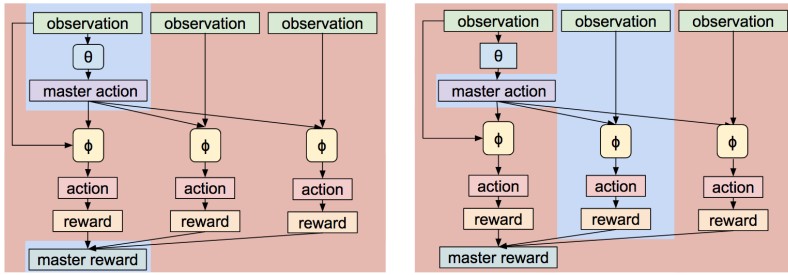

Figure 2: Unrolled structure for a master policy action lasting $N = 3$ timesteps. Left: When training the master policy, the update only depends on the master policy's action and total reward (blue region), treating the individual actions and rewards as part of the environment transition (red region). Right: When training sub-policies, the update considers the master policy's action as part of the observation (blue region), ignoring actions in other timesteps (red region)

## 5 RATIONALE

We will now provide intuition for why this framework leads to a set of sub-policies $\phi$ which allow agents to quickly reach high reward when learning $\theta$ on a new task. In metalearning methods, it is common to optimize for reward over an entire inner loop (in the case of MLSH, training $\theta$ for $T$ iterations). However, we instead choose to optimize $\phi$ towards maximizing reward within a single episode. Our argument relies on the assumption that the warmup period of $\theta$ will learn an optimal master policy, given a set of fixed sub-polices $\phi$. As such, the optimal $\phi$ at $\theta_{\text{final}}$ is equivalent to the optimal $\phi$ for training $\theta$ from scratch. While this assumption is at some times false, such as when a gradient update overshoots the optimal $\theta$ policy, we empirically find the assumption accurate enough for training purposes.

Next, we consider the inclusion of a warmup period. It is important that $\phi$ only be updated when $\theta$ is at a near-optimal level. A motivating example for this is a navigation task containing two possible destinations, as well as two sub-policies. If $\theta$ is random, the optimal sub-policies both lead the agent to the midpoint of the destinations. If $\theta$ contains information on the correct destination, the optimal sub-policies consist of one leading to the first destination, and the other to the second.

Finally, we will address the reasoning behind limiting the update period to $U$ iterations. As we update the sub-policy parameters $\phi$ while reusing master policy parameters $\theta$, we are assuming that re-training $\theta$ will result in roughly the same master policy. However, as $\phi$ changes, this assumption holds less weight. We therefore stop and re-train $\theta$ once a threshold of $U$ iterations has passed.

## 6 EXPERIMENTS

We hypothesize that meaningful sub-policies can be learned by operating over distributions of tasks, in an efficient enough manner to handle complex physics domains. We also hypothesize that sub-policies can be transferred to complicated tasks outside the training distribution. In the following section, we present a series of experiments designed to test the performance of our method, through comparison to baselines and past methods with hierarchy.

### 6.1 EXPERIMENTAL SETUP

We present a series of environments containing both shared and task-specific information. We examine two curves: the overall learning on the entire distribution ($\phi$), as well as the learning on a sampled individual task ($\theta$). For overall training, we compare to a baseline of a shared policy trained jointly across all tasks from the distribution. We also compare to running MLSH without a warmup period.

In the sampled individual task experiments, our MLSH agent utilizes sub-policies ($\phi$) previously trained on the entire distribution, and only updates the master policy ($\theta$) towards the new task. To test the importance of the sub-policy structure, we compare against fine-tuning a single policy that has been optimized across all tasks. We also compare against training a new single policy from scratch, to test if the learned sub-policies are useful.

For both master and sub-policies, we use 2 layer MLPs with a hidden size of 64. Master policy actions are sampled through a softmax distribution. We train both master and sub-policies using policy gradient methods, specifically PPO (Schulman et al., 2017). For collecting experience, we compute a batchsize of $D$=2000 timesteps. We use a much larger learning rate for $\theta$ (0.01) than for $\phi$ (0.0003), since $\phi$ parameters should remain relatively consistent throughout a single warmup and joint-update period.

While the base MLSH algorithm is sequential, we run all experiments in a parallel multi-core setup for faster wall-clock training time. We split 120 cores into 10 groups of 12 cores, where a group represents a single MLSH learner which uses 12 cores to to collect experience in parallel. All groups sample individual tasks from the task distribution, and only $\phi$ parameters are shared. Viewed as a whole, we are optimizing a shared set of $\phi$ parameters towards 10 sampled tasks in parallel.

To prevent periods where the $\phi$ parameters are receiving no gradients, we stagger the warmup periods of each group, so a new group enters warmup as soon as another group leaves. Once a group has finished both its warmup and joint-update period, a new task is sampled along with a new random

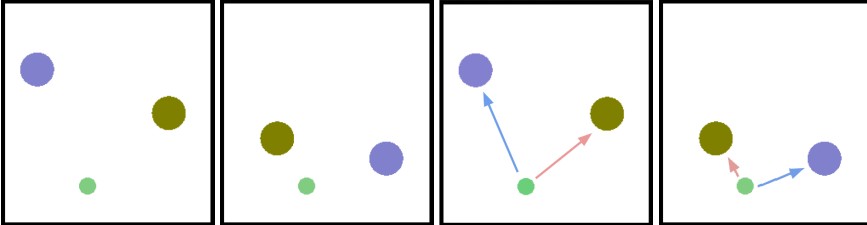

Figure 3: Sampled tasks from 2D moving bandits. Small green dot represents the agent, while blue and yellow dots represent potential goal points. Right: Blue/red arrows correspond to movements when taking sub-policies 1 and 2 respectively.

initialization of $\theta$, both of which are shared within all cores in the group. Warmup and joint-update lengths for individual environment distributions will be described in the following section. As a general rule, a good warmup duration represents the amount of gradient updates required to approach convergence of $\theta$.

## 6.2 CAN MEANINGFUL SUB-POLICIES BE LEARNED OVER A DISTRIBUTION OF TASKS, AND DO THEY OUTPERFORM A SHARED POLICY?

Our motivating problem is a 2D moving bandits task (Figure 3), in which an agent is placed in a world and shown the positions of two randomly placed points. The agent may take discrete actions to move in the four cardinal directions, or opt to stay still. One of the two points is marked as correct, although the agent does not receive information on which one it is. The agent receives a reward of 1 if it is within a certain distance of the correct point, and a reward of 0 otherwise. Each episode lasts 50 timesteps, and master policy actions last for 10. We use two sub-policies, a warmup duration of 9, and a joint-update duration of 1.

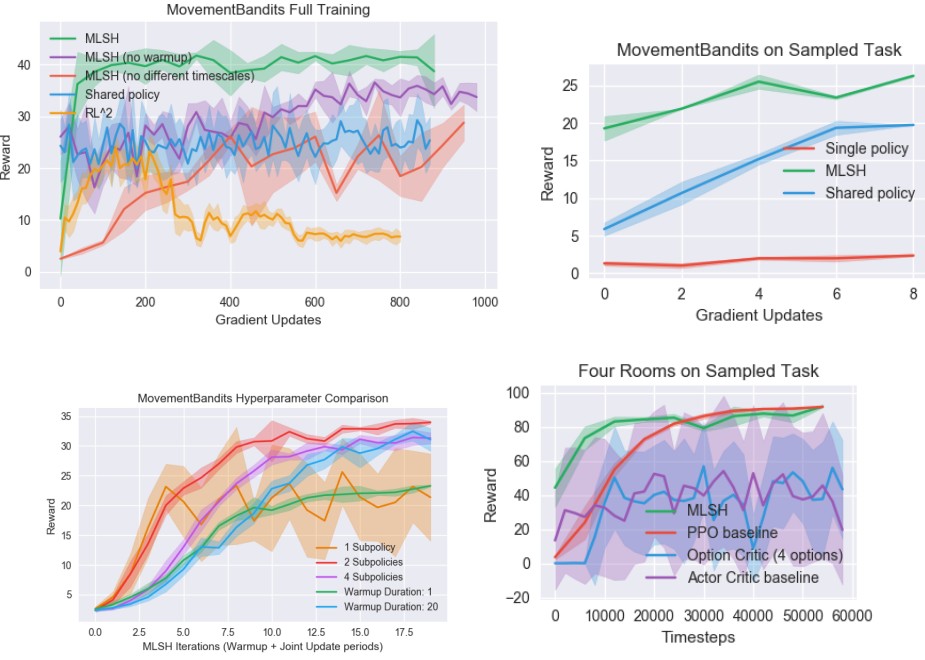

Figure 4: Learning curves for 2D Moving Bandits and Four Rooms.

After training, MLSH learns sub-policies corresponding to movement towards each potential goal point. Training a master policy is faster than training a single policy from scratch, as we are tasked

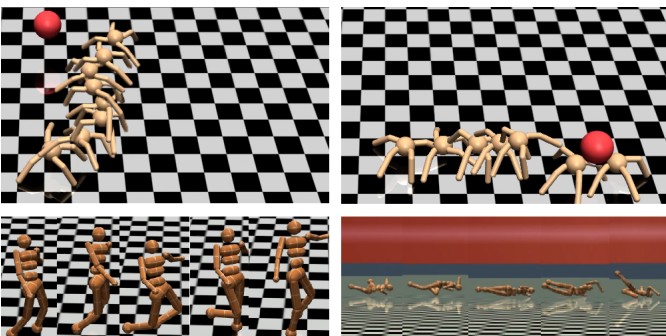

Figure 5: Top: Ant Twowalk. Ant must maneuver towards red goal point, either towards the top or towards the right. Bottom Left: Walking. Humanoid must move horizontally while maintaining an upright stance. Bottom Right: Crawling. Humanoid must move horizontally while a height-limiting obstacle is present.

only with discovering the correct goal, rather than also learning primitive movement. Learning a shared policy, on the other hand, results in an agent that always moves towards a certain goal point, ignoring the other and thereby cutting expected reward by half. We additionally compare to an $RL^2$ policy (Duan et al., 2016), which encounters the same problem as the shared policy and ignores one of the goal points.

We perform several ablation tests within the 2D moving bandits task. Removing the warmup period results in an MLSH agent which at first has both sub-policies moving to the same goal point, but gradually shifts one sub-policy towards the other point. Running the master policy on the same timescale as the sub-policies results in similar behavior to simply learning a shared policy, showing that the temporal extension of sub-policies is key. Finally, we run a hyperparameter comparison to test the influence of the sub-policy count and warmup duration.

### 6.3 HOW DOES MLSH COMPARE TO PAST METHODS IN THE HIERARCHICAL DOMAIN?

To compare to past methods, we consider the four-rooms domain described in Sutton et al. (1999) and expanded in Option Critic (Bacon et al., 2016). The agent starts at a specific spot in the grid-world, and is randomly assigned a goal position. A reward of 1 is awarded for being in the goal state. Episodes last for 100 timesteps, and master policy actions last for 25. We utilize four sub-policies, a warmup time of 20, and a joint-update time of 30.

First, we repeatedly train MLSH and Option Critic on many random goals in the four-rooms domain, until reward stops improving. Then, we sample an unseen goal position and fine-tune. We compare against baselines of training a single policy from scratch, using PPO against MLSH, and Actor Critic against Option Critic. In Figure 4, while Option Critic performs similarly to its baseline, we can see MLSH reach high reward faster than the PPO baseline. It is worth noting that when fine-tuning, the PPO baseline naturally reaches more stable reward than Actor Critic, so we do not compare MLSH and Option Critic directly.

### 6.4 IS THE MLSH FRAMEWORK SAMPLE-EFFICIENT ENOUGH TO LEARN DIVERSE SUB-POLICIES IN PHYSICS ENVIRONMENTS?

To test the scalability of the MLSH algorithm, we present a series of physics-based tasks which we describe below, all which are simulated through Mujoco (Todorov et al., 2012). Diverse sub-policies are naturally discovered, as shown in Figure 5 and Figure 6. Episodes last 1000 timesteps, and master policy actions last 200. We use a warmup time of 20, and a joint-update time of 40.

In the Twowalk tasks, we would like to examine if simulated robots can learn directional movement primitives. We test performance on a standard simulated four-legged ant, and use a sub-policy count of two. A destination point is placed in either the top edge of the world or the right edge of the world. Reward is given based on negative distance to this destination point.

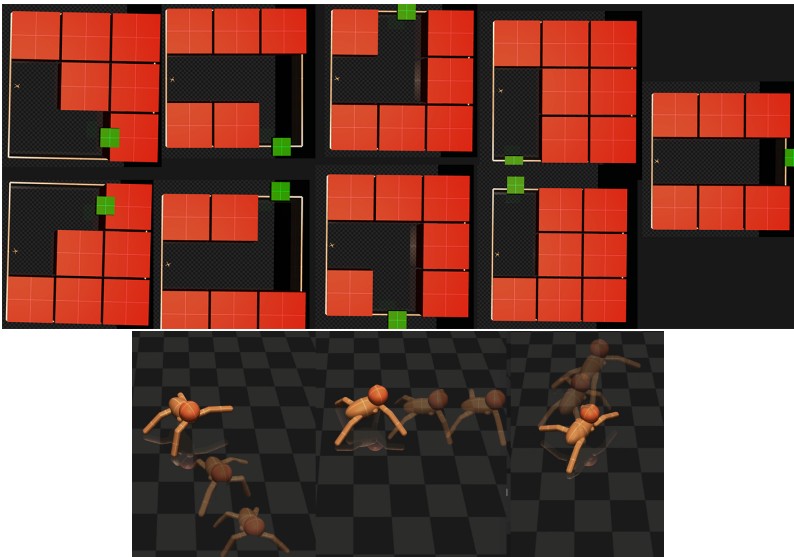

Figure 6: Top: Distribution of mazes. Red blocks are impassable tiles, and green blocks represent the goal. Bottom: Sub-policies learned from mazes to move up, right, and down.

In addition, we would like to determine if diverse sub-policies can be automatically discovered solely through interaction with the environment. We present a task where Ant robots must move to destination points in a set of mazes (Figure 6). Without human supervision, Ant robots are able to learn directional movement sub-policies in three directions, and use them in combination to solve the mazes.

In the Walk/Crawl task, we would like to determine if Humanoid robots can learn a variety of movement styles. Out of two possible locomotion objectives, one is randomly selected. In the first objective, the agent must move forwards while maintaining an upright stance. This was designed with a walking behavior in mind. In the second objective, the agent must move backwards underneath an obstacle limiting vertical height. This was designed to encourage a crawling behavior.

Additionally, we test the transfer capabilities of sub-policies trained in the Walk/Crawl task by introducing an unseen combination task. The Humanoid agent must first walk forwards until a certain distance, at which point it must switch movements, turn around, and crawl backwards under an obstacle.

| Reward on Walk/Crawl combination task | |
| --- | --- |
| MLSH Transfer | 14333 |
| Shared Policy Transfer | 6055 |
| Single Policy | -643 |

On both Twowalk and Walk/Crawl tasks, MLSH significantly outperforms baselines, displaying scalability into complex physics domains. Ant robots learn temporally-extended directional movement primitives that lead to efficient exploration of mazes. In addition, we successfully discover diverse Humanoid sub-policies for both walking and crawling.

## 6.5 CAN SUB-POLICIES BE USED TO LEARN IN AN OTHERWISE UNSOLVABLE SPARSE PHYSICS ENVIRONMENT?

Finally, we present a complex task that is unsolvable with naive PPO. The agent controls an Ant robot which has been placed into an obstacle course. The agent must navigate from the bottom-left corner to the top-right corner, to receive a reward of 1. In all other cases, the agent receives a reward of 0. Along the way, there are obstacles such as walls and a chasing enemy. We periodically reset the joints of the Ant robot to prevent it from falling over. An episode lasts for 2000 timesteps, and

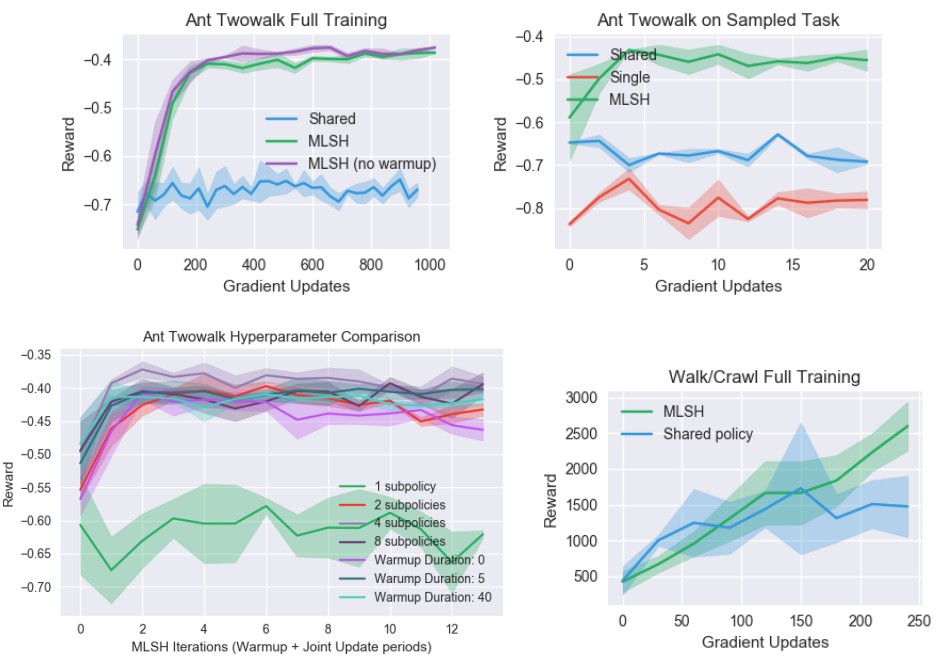

Figure 7: Learning curves for Twowalk and Walk/Crawl tasks

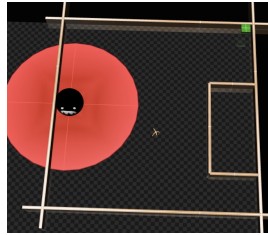

Figure 8: Ant Obstacle course task. Agent must navigate to the green square in the top right corner. Entering the red circle causes an enemy to attack the agent, knocking it back.

master policy actions last 200. To solve this task, we use sub-policies learned in the Ant Twowalk tasks. We then fine-tune the master policy on the obstacle course task.

In the sparse reward setting, naive PPO cannot learn, as exploration over the space of primitive action sequences is unlikely to result in reward signal. On the other hand, MLSH allows for exploration over the space of sub-policies, where it is easier to discover a sequence that leads to reward.

| Reward on Ant Obstacle task | |
| --- | --- |
| MLSH Transfer | 193 |
| Single Policy | 0 |

# 7 DISCUSSION

In this work, we formulate an approach for the end-to-end metalearning of hierarchical policies. We present a model for representing shared information as a set of sub-policies. We then provide a framework for training these models over distributions of environments. Even though we do not optimize towards the true objective, we achieve significant speedups in learning. In addition, we naturally discover diverse sub-policies without the need for hand engineering.

## 7.1 FUTURE WORK

As there is no gradient signal being passed between the master and sub-policies, the MLSH model utilizes hard one-hot communication, as opposed to methods such as Gumbel-Softmax (Jang et al., 2016). This lack of a gradient also allows MLSH to be learning-method agnostic. While we used policy gradients in our experiments, it is entirely feasible to have the master or sub-policies be trained with evolution (Eigen) or Q-learning (Watkins & Dayan, 1992).

From another point of view, our training framework can be seen as a method of joint optimization over two sets of parameters. This framework can be applied to other scenarios than learning sub-policies. For example, distributions of tasks with similar observation distributions but different reward functions could be solved with a shared observational network, while learning independent policies.

This work draws inspiration from the domains of both hierarchical reinforcement learning and meta-learning, the intersection at which architecture space has yet to be explored. For example, the set of sub-policies could be condensed into a single neural network, which receives a continuous vector from the master policy. If sample efficiency issues are addressed, several approximations in the MLSH method could be removed for a more unbiased estimator – such as training $\phi$ to maximize reward on the entire $T$-timesteps, rather than on a single episode. We believe this work opens up many directions in training agents that can quickly adapt to new tasks.

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
