# OpenReview forum: "META LEARNING SHARED HIERARCHIES"
_ICLR.cc/2018/Conference — Accept (Poster)_

### Official Review · AnonReviewer1 · 2017-11-27
**Metalearning with shared hierarchies**

**Rating:** 4
**Confidence:** 4

**Review:**

This paper considers the reinforcement learning problem setup in which an agent must solve not one, but a set of tasks in some domain, in which the state space and action space are fixed. The authors consider the problem of learning a useful set of ‘sub policies’ that can be shared between tasks so as to jump start learning on new tasks drawn from the task distribution.

I found the paper to be generally well written and the key ideas easy to understand on first pass. The authors should be commended for this. Aside from a few minor grammatical issues (e.g. missing articles here and there), the writing cannot be too strongly faulted.

The problem setup is of general interest to the community. Metalearning in the multitask setup seems to be gaining attention and is certainly a necessary  step towards building rapidly adaptable agents.

While the concepts were clearly introduced, I think the authors need to make, much more strongly, the case that the method is actually valuable. In that vein, I would have liked to see more work done on elucidating how this method works ‘under the hood’. For example, it is not at all clear how the number of sub policies affects performance (one would imagine that there is a clear trade off), nor how this number should be chosen. It seems obvious that this choice would also affect the subtle dynamics between holding the master policy constant while updating the sub policies and vice versa. While the authors briefly touch on some of these issues in the rationale section, I found these arguments largely unsubstantiated. Moreover, this leads to a number of unjustified hyper-parameters in the method which I suspect would affect the training catastrophically without significant fine-tuning.

There are also obvious avenues to be followed to check/bolster the intuitions behind the method. By way of example, my sense is that the procedure described in the paper uncovers a set of sub policies that form a `good’ cover for the task space - if so simply plotting out what they policies look like (or better yet how they adapt in time) would be very insightful (the rooms domain is perhaps a good candidate for this).

While the key ideas are clearly articulated  the practical value of the procedure is insufficiently motivated. The paper would benefit hugely from additional analysis.

---

> ### Author Response · Authors · 2017-12-12
> **Analysis on Parameter Choice**
>
> Hey, appreciate the feedback.
>
> To address your concern about how performance depends on hyperparameters, we ran additional experiments comparing the effects of various parameter adjustments. See the graph (https://imgur.com/a/TLyQv), which we have added in Fig.9 on the current revision. (Default parameters are a sub-policy count of 2, and a warmup duration of 20). As long as a few minimums are met (at least 2 sub-policies), performance is not overly dependent on fine-tuned parameters. The parameters we describe in the paper can be seen as a “baseline minimum” of parameters to reach a strong solution on the various tasks.
>
> Regarding displaying the behavior of sub-policies, we show a decomposition of the three sub-policies discovered in the Maze task in Figure 6: moving up, right, and down. We display how the policies adapt over time in our supplemental videos, linked on the first page (https://sites.google.com/site/mlshsupplementals/, specifically https://www.youtube.com/watch?v=9nvjy9aJi50).

---

### Official Review · AnonReviewer2 · 2017-11-27
**very vague paper**

**Rating:** 6
**Confidence:** 3

**Review:**

Please see my detailed comments in the "official comment"

The extensive revisions addressed most of my concerns

Quality
======
The idea is interesting, the theory is hand-wavy at best (ADDRESSED but still a bit vague), the experiments show that it works but don't evaluate many interesting/relevant aspects (ADDRESSED). It is also unclear how much tuning is involved (ADDRESSED).

Clarity
=====
The paper reads OK. The general idea is clear but the algorithm is only provided in vague text form (and actually changing from sequential to asynchronous without any justification why this should work) (ADDRESSED) leaving many details up the the reader's best guess (ADDRESSED).

Originality
=========
The idea looks original.

Significance
==========
If it works as advertised this approach would mean a drastic speedup on previously unseen task from the same distribution.

Pros and Cons
============
+ interesting idea
- we do everything asynchronously and in parallel and it magically works (ADDRESSED)
- many open questions / missing details (ADDRESSED)

---

> ### Author Response · Authors · 2017-12-12
> **Addressed Clarifications**
>
> Hey, thanks for the feedback. We’ve addressed some clarifications in the response to your official comment, titled "Proposed Changes". We hope that these ideas clear up misunderstandings, and fill in details that may have been explained unclearly.

---

### Official Review · AnonReviewer3 · 2017-11-28
**This paper proposes a novel method for inducing temporal hierarchical structure in a specialized multi-task setting.**

**Rating:** 7
**Confidence:** 3

**Review:**

This paper proposes a novel hierarchical reinforcement learning method for a fairly particular setting.  The setting is one where the agent must solve some task for many episodes in a sequence, after which the task will change and the process repeats.  The proposed solution method splits the agent into two components, a master policy which is reset to random initial weights for each new task, and several sub-policies (motor primitives) that are selected between by the master policy every N steps and whose weights are not reset on task switches.  The core idea is that the master policy is given a relatively easy learning task of selecting between useful motor primitives and this can be efficiently learned from scratch on each new task, whereas learning the motor primitives occurs slowly over many different tasks.  To push this motivation into the learning process, the master policy is updated always but the sub-policies are only updated after an extended warmup period (called the joint-update or training period).  This experiments include both small domains (moving to 2D goals and four-rooms) and more complex physics simulations (4-legged ants and humanoids).  In both the simple and complex domains, the proposed method (MLSH) is able to robustly achieve good performance.

This approach to obtaining complex structured behavior appears impressive despite the amount of temporal structure that must be provided to the method (the choice of N, the warmup period, and the joint-update period).  Relying on the temporal structure for the hierarchy, and forcing the master policy to be relearned from scratch for each new task may be problematic in general, but this work shows that in some complex settings, a simple temporal decomposition may be sufficient to encourage the development of reusable motor primitives and to also enable quick learning of meta-policies over these motor-primitives.  Moreover, the results show that these temporal hierarchies are helpful in these domains, as the corresponding non-hierarchical methods failed on the more challenging tasks.

The paper could be improved in some places (e.g. unclear aliases of joint-update or training periods, describing how the parameters were chosen, and describing what kinds of sub-policies are learned in these domains when different parameter choices are made).

---

> ### Author Response · Authors · 2017-12-12
> **Thanks for the response**
>
> Thanks for the response. We’ll fix the typo of “training period”-> “joint-update period” in the next version. We’ll also clean up the intuition behind parameter choice (see our response “Analysis on Parameter Choice”).

---

### Comment · AnonReviewer2 · 2017-11-27
**Please improve algorithm description**

The paper proposes to learn sub-policies (or motor primitives, options, etc.) jointly with a higher level policy by optimizing for multiple tasks simultaneously

- Algorithm 1 makes it sound like all the learning happens sequentially, later on we learn that a multi-core setup is used. The details on this remain very vague. As far as I can guess we have 10 groups (with 12 cores each). All cores within a group are assigned the exact same task (shared theta), the parameters phi for the sub-policies is shared between all nodes. The text reads like the parameters theta are forgotten and the learning thereof is restarted with the exact same task. Hence we optimize for exactly 10 tasks in parallel?

- Is the number of sup-policies pre-defined/hard-coded/hand-designed? What happens if you have too many/not enough?

- The argumentation in Sect. 5 is vague and holds only if the tasks are learned sequentially. To me it sounds like you need to ensure that you don't update phi too much, otherwise it might unlearn something useful for the previous tasks. In the 2D moving bandit problem this seems to be achieved by only updating phi for a small amount of time.

- I am not convinced that the above is solved by staggering the tasks in the asynchronous setting. While still in the warm-up phase (i.e,. learning theta) the agents associated to a certain task need to cope with the fact that the phi is changed simultaneously, hence they have to play catch-up with the changing representation while trying to improve their performance.

- Another interesting experiment would be to test how much the system unlearns, e.g., by optimizing for a task, switching to a few other tasks, freezing phi and testing if the first task can still achieve the same performance

- The plots Fig. 4/7 are a bit unclear. My guess is "full training" means learning from scratch as described in Sect. 6.1, "sampled tasks" means trying whether the learned sub-policies also work for a previously unseen task. Here again the question: What happens if you freeze phi? How well do phi updated on the new tasks work on the original ones? Related question: Why is there no plot on the combination task (Fig. 7) and full training on Four Rooms?

- Sect. 6.4 "series of tasks" is a bit unclear

- Sect. 6.4: Why is the ratio of warm-up and training so different compared to the 2D bandits? How much influence does this parameter have on the performance of the approach?

---

> ### Author Response · Authors · 2017-12-01
> **Proposed Changes**
>
> Thanks for taking the time to review and give feedback. We’ve addressed the main points and proposed some changes in the next version to clear up explanations and reasoning.
>
> > Algorithm 1 makes it sound like...parallel?
>
> While the core algorithm can be run sequentially, we use a multi-core setup in experiments to speed up the process. There may be some confusion in the description -- after the group resets theta, a new task is sampled, and all cores learn on this same new task. Therefore at any given time we are optimizing for 10 tasks in parallel, but these tasks are constantly being re-sampled from the distribution. We will clarify this in the next revision.
>
> > Is the number of sup-policies pre-defined/hard-coded/hand-designed? What happens if you have too many/not enough?
>
> For simplicity’s sake, we pre-define the number of sub-policies, treating it as a hyperparameter. In the Future Work section, we describe a potential method for condensing multiple sub-policies into a single network, allowing the agent to learn any distribution of sub-policies.
>
> With a small number of sub-policies, the agent may be less robust to new tasks (as it learns fewer behaviors). With a large number of sub-policies, it takes longer to train agents.
>
> > The argumentation in Sect. 5 is vague...time.
>
> The point about not updating phi too much is correct (we address it below). A key point in the Sect.5 argument is that sub-policies should only be trained in conjunction with a strong master policy, which is the rationale behind the warmup period.
>
> > I am not convinced that the above is solved by staggering the tasks in the asynchronous setting...
>
> In practice, we use a small phi learning rate (0.0003) compared to the theta learning rate (0.01), as defined in the 6.1 Experimental Setup. Our goal here is that small changes in the representation (phi) are negligible in the short-run training of theta, but will build up in the long-run.
>
> We’ll add in this reasoning behind the learning-rate choices in the next revision.
>
> > Another interesting experiment would be to test how much the system unlearns, e.g., by optimizing for a task, switching to a few other tasks, freezing phi and testing if the first task can still achieve the same performance
> > The plots Fig. 4/7 are a bit unclear. My guess is "full training" means learning from scratch as described in Sect. 6.1, "sampled tasks" means trying whether the learned sub-policies also work for a previously unseen task. Here again the question: What happens if you freeze phi? How well do phi updated on the new tasks work on the original ones? Related question: Why is there no plot on the combination task (Fig. 7) and full training on Four Rooms?
>
> You’re correct on the meaning of “full training” and “sampled tasks”. We’ll add a description in the caption to clear things up.
>
> Regarding the freeze phi experiment: When running the “Sampled Task” experiments, only theta is trained, so phi is frozen. If phi was overfitting/unlearning on every new task, the agent would perform poorly on an unseen “Sampled Task”.
>
> The plot for the combination task is uninteresting since the rewards are so different. The different trials (MLSH Transfer, Shared Policy Transfer, Single Policy) never pass each other in performance.
>
> On four rooms, we don’t include a full training since the base methods compared (PPO and Actor Critic) have vastly different sample efficiencies. Instead, we just train until both baselines have reached convergence.
>
> > Sect. 6.4 "series of tasks" is a bit unclear
>
> Thanks -- we’ll clarify this to “series of tasks involving robotic locomotion in the physics domain”
>
> > Sect. 6.4: Why is the ratio of warm-up and training so different compared to the 2D bandits? How much influence does this parameter have on the performance of the approach?
>
> The physics domain has a more complicated learning task for sub-policies compared to the 2D task, so training is naturally slower. However, master policies have the same learning task in both situations (select a sub-policy). So we give more training updates per warmup in the physics task. While it’s important to have a warmup period (as shown in Fig 4, MLSH performs worse when not including a warmup), the ratio doesn’t need to be precise. It’s always more accurate to have a long warmup period and short training period, but the agent will take longer to train. We’ll add this intuition in the next revision.

---

> > ### Author Response · Authors · 2017-12-12
> > **Update on Parameter Effects**
> >
> > We've run experiments comparing the effects of various parameters (sub-policy count, warmup ratio). See (https://imgur.com/a/TLyQv), and our comment "Analysis on Parameter Choice" for more details.

---

> > > ### Comment · AnonReviewer2 · 2017-12-12
> > > **Warmup Duration doesn't have any influence?**
> > >
> > > Thanks for the extensive replies and clarifications!
> > > In the new plot it almost looks like the warmup does not have any influence at all. So it might actually be highly task dependent whether it is needed/helpful/detrimental or not...

---

> > > > ### Author Response · Authors · 2017-12-14
> > > > **Analysis on Additional Task**
> > > >
> > > > Point taken. We've run another set of hyperparameter experiments on the MovementBandit task. See graph (https://imgur.com/a/D7YMx), which we will add in the next paper revision. (Default is 2 subpolicies, warmup duration of 10).
> > > >
> > > > For sure, the influence of the warmup ratio depends on the task. However, as a rule of thumb, a long warmup period (or large subpolicy count) will simply result in a longer training time, rather than a downgrade in final performance.
> > > >
> > > > In the MovementBandit parameter comparison, performance is only drastically lowered if the warmup duration is close to zero or the subpolicy count is one. On the other hand, when the warmup duration is 40 or the subpolicy count is 4, the agent still converges to the optimal solution, albeit at a slightly slower pace.

---

### Public Comment · (anonymous) · 2017-11-29
**Unclear about novelty**

The work is interesting but it is unclear how novel it is. There exists similar work that learns something very similar to the hierarchy in this paper.

X. B. Peng, G. Berseth, and M. Van de Panne. 2016. Terrain-Adaptive Locomotion Skills Using Deep Reinforcement Learning. ACM Transactions on Graphics (Proc. SIGGRAPH 2016) 35, 5.

However, in this previous work the sub-policy is not a neural network but having the sub-policy not be a neural network is not a novel idea.

---

> ### Author Response · Authors · 2017-12-01
> **Clarification**
>
> The idea of learning a hierarchy of sub-policies has been explored in past work, many of which we cite and discuss in Section 2:
>
> Pierre-Luc Bacon, Jean Harb, and Doina Precup. The option-critic architecture. arXiv preprint arXiv:1609.05140, 2016.
>
> Carlos Florensa, Yan Duan, and Pieter Abbeel. Stochastic neural networks for hierarchical reinforcement learning. In International Conference on Learning Representations, 2017.
>
> Richard S Sutton, Doina Precup, , and Satinder Singh. Between mdps and semi-mdps: A framework for temporal abstraction in reinforcement learning. In Artificial intelligence, 1999.
>
> In contrast to many previous works, our method aims to learn sub-policies automatically, without the need for hand engineering (in the paper you mentioned, they design running and leaping policies). In addition, we focus on the idea of sharing sub-policies over distributions of tasks, rather than on single tasks.

---

### Public Comment · (anonymous) · 2017-12-15
**Issues with the released code**

Hi,

Interesting paper and thanks for releasing the code! But can you please make it working and take case about some of the reported issues? Thanks!

---

> ### Author Response · Authors · 2017-12-16
> **Re-structure of repo**
>
> Hey, the repo has been restructured so it should be easier to install correctly.

---

### Author Response · Authors · 2018-01-05
**Changes in latest revision**

In the latest paper revision we have added the following fixes/clarifications:

- Added graphs showing a hyperparameter comparison on sub-policy count and warmup-duration for MovementBandits and Ant-Twowalk tasks.
- Clarified the details of the multi-core training process in 6.1 "Experimental Setup".
- Added reasoning behind the learning rates of theta and phi.
- Added details and reasoning behind baseline comparisons for "Sampled Task" experiments.
- Changed  "training period" to "joint-update period" for consistency.

---

### Decision · Program_Chairs · 2018-01-29
**ICLR 2018 Conference Acceptance Decision**

**Decision:**

Accept (Poster)

**Comment:**

This paper presents a fairly straightforward algorithm for learning a set of sub-controllers that can be re-used between tasks.  The development of these concepts in a relatively clear way is a nice contribution.  However, the real problem is how niche the setup is.  However, it's over the bar in general.